# Glucokinase Inactivation Ameliorates Lipid Accumulation and Exerts Favorable Effects on Lipid Metabolism in Hepatocytes

**DOI:** 10.3390/ijms24054315

**Published:** 2023-02-21

**Authors:** Ziyan Xie, Ting Xie, Jieying Liu, Qian Zhang, Xinhua Xiao

**Affiliations:** 1Key Laboratory of Endocrinology, Ministry of Health, Department of Endocrinology, Peking Union Medical College Hospital, Chinese Academy of Medical Sciences & Peking Union Medical College, Beijing 100730, China; 2Department of Medical Research Center, Peking Union Medical College Hospital, Chinese Academy of Medical Sciences & Peking Union Medical College, Beijing 100730, China

**Keywords:** glucokinase, GCK-MODY, hepatic lipid metabolism, inflammation, lipotoxicity

## Abstract

Glucokinase-maturity onset diabetes of the young (GCK-MODY) is a kind of rare diabetes with low incidence of vascular complications caused by *GCK* gene inactivation. This study aimed to investigate the effects of GCK inactivation on hepatic lipid metabolism and inflammation, providing evidence for the cardioprotective mechanism in GCK-MODY. We enrolled GCK-MODY, type 1 and 2 diabetes patients to analyze their lipid profiles, and found that GCK-MODY individuals exhibited cardioprotective lipid profile with lower triacylglycerol and elevated HDL-c. To further explore the effects of GCK inactivation on hepatic lipid metabolism, GCK knockdown HepG2 and AML-12 cell models were established, and in vitro studies showed that GCK knockdown alleviated lipid accumulation and decreased the expression of inflammation-related genes under fatty acid treatment. Lipidomic analysis indicated that the partial inhibition of GCK altered the levels of several lipid species with decreased saturated fatty acids and glycerolipids including triacylglycerol and diacylglycerol, and increased phosphatidylcholine in HepG2 cells. The hepatic lipid metabolism altered by GCK inactivation was regulated by the enzymes involved in de novo lipogenesis, lipolysis, fatty acid β-oxidation and the Kennedy pathway. Finally, we concluded that partial inactivation of GCK exhibited beneficial effects in hepatic lipid metabolism and inflammation, which potentially underlies the protective lipid profile and low cardiovascular risks in GCK-MODY patients.

## 1. Introduction

Glucokinase (GCK) catalyzes the phosphorylation of glucose to glucose 6-phosphate and is generally considered the initial glucose-sensing component and gatekeeper for glucose metabolism. The gene expression and protein level of GCK are enriched in the pancreas, liver, intestine, hypothalamus and pituitary [1]. In pancreatic β-cells, GCK participates in the regulation of glucose-induced insulin secretion. In the liver, GCK plays a leading role in glycogen synthesis and glycolysis [2].

Due to its important role in glucose homeostasis, the loss of function of GCK leads to diseases. Glucokinase-maturity-onset diabetes of the young (GCK-MODY) is caused by heterozygous inactivating mutations in GCK and impaired glucose sensing. However, unlike type 1 and 2 diabetes (T1D and T2D) or other MODYs, patients with GCK-MODY generally have a favorable prognosis without the requirement of antidiabetic treatment [3]. Additionally, GCK-MODY patients rarely suffer cardiovascular complications with the same risks as nondiabetic healthy individuals [4,5]. The low occurrence of vascular complications in GCK-MODY makes it a natural model for investigating the protective mechanisms of cardiovascular disorders under prolonged hyperglycemia. Research has shown that GCK-MODY individuals exhibit favorable serum lipid, with lower levels of triacylglycerols (TAGs) and higher high-density lipoproteins (HDLs), even compared with healthy subjects [6]. Our previous work [7] demonstrated that compared to T2D, several serum phosphatidylcholines (PCs) and plasmalogen PCs (PCps) were significantly increased in GCK-MODY, which contribute to the antiapoptotic and anti-inflammatory effects of HDL. Furthermore, evidence has suggested that the distinct lipid profile of GCK-MODY individuals exerts cardioprotective effects [8]. On the other hand, GCK activators (GKAs) have been reported to cause adverse effects including hyperlipidemia, hepatic fat accumulation and hepatic steatosis, in addition to hypoglycemic effects in both clinical trials [9,10,11] and animal studies [12,13], indicating the potential roles of GCK in maintaining lipid homeostasis.

Overall, current observations illustrated that in addition to glucose homeostasis, GCK also plays a crucial role in regulating lipid metabolism, and the inactivation of GCK may underlie the antiatherogenic profile associated with GCK-MODY. However, the association between GCK mutations and lipid profile and its underlying mechanism remains undefined. Given the critical role of the liver in lipid homeostasis of the body and the relatively high expression of GCK in the liver, we speculate that partial inactivation of GCK could exert favorable effects on hepatic lipid metabolism, probably through regulating key enzymes involved in metabolism pathways, thereby contributing to the cardioprotective lipid profile of GCK-MODY. The objective of the present study was to explore the protective lipid profile in GCK-MODY patients compared with T1D and T2D and the effects of GCK knockdown on hepatic lipid accumulation and inflammation in cell models.

## 2. Results

### 2.1. GCK-MODY Patients Exert Favorable Lipid Profile

The characteristics and lipid profile of GCK-MODY, T1D and T2D patients and nondiabetic control subjects are shown in Table 1. In accordance with the type of diabetes, the glucose profile including FBG (*p* < 0.0001), HbA1c (*p* < 0.001) and GA (*p* < 0.0001) were increasingly elevated in three patient groups. The lipid metabolic profiles of GCK-MODY were significantly improved compared to T1D and T2D and were comparable to the normal control. The levels of TAG (*p* < 0.0001), TC (*p* < 0.0001) and LDL-c (*p* < 0.0001) were remarkably decreased in GCK-MODY compared with T2D. Additionally, a significant elevation in HDL-c was also shown in GCK-MODY compared with both T1D (*p* = 0.0060) and T2D (*p* < 0.0001). Furthermore, the level of CRP was also lower in GCK-MODY than T1D (*p* = 0.0256) and T2D (*p* = 0.0168), indicating a reduced cardiovascular risk.

### 2.2. GCK Knockdown Improved Lipid Accumulation in HFA-Treated HepG2 Cells

As the liver is the central organ of lipid metabolism in the body, we examined the effects of GCK inactivation on hepatic lipid metabolism via establishing in vitro liver cell models to explore the possible mechanism of the unique lipid profile in GCK-MODY individuals. Lentivirus transfection was applied to generate stable GCK knockdown in the human HpeG2 cell line. Glucokinase activity determination and Western blot were used to validate the transfection efficacy. The glucokinase activity was significantly reduced by 50% in GCK knockdown HepG2 cells (*p* < 0.0001) (Figure 1A). Consistently, the level of GCK protein also displayed remarkable downregulation (*p* = 0.0017) (Figure 1B).

To investigate the impacts of GCK inactivation on hepatic lipid metabolism, the HepG2 cells were challenged with HFA to induce lipotoxicity. Oil Red O staining suggested that the knockdown of GCK significantly alleviated the HFA-treated lipid accumulation of HepG2 cells (Figure 1C). Meanwhile, the intracellular TAG content was also significantly reduced in the GCK knockdown group under HFA challenge (*p* < 0.0001) (Figure 1D). These results indicated that GCK knockdown reduced TAG content and prevented lipid accumulation in HFA-treated HepG2 cells.

### 2.3. Lipid Profile in GCK Knockdown HepG2 Cells

Lipidomic analysis was applied to reveal the entire lipid content variation caused by GCK knockdown (Figure 2). The score scatter plot of OPLS-DA (Figure 2A) showed that the samples from the control and GCK knockdown groups were independently grouped in both negative and positive ionization modes, which implied that the partial inactivation of GCK resulted in significant changes in the lipidome in HepG2 cells. Hierarchical cluster analysis was also performed on the screened differential metabolites, based on the threshold of variable importance values (VIP > 1.0) and *p* values (<0.05) (Appendix A).

The top 30 lipid metabolites were selected for further analysis based on the VIP score. The statistical data are depicted in a heatmap (Figure 2B). The selected metabolites could be classified as five lipid species, including glycerophospholipid, glycerolipid, sphingolipid, acylcarnitine and glycolipid. The bar chart shows the relative differences in the lipid species in the GCK knockdown group compared with the control (Figure 2C). The abundance of the identified lipids displayed a significant increase in PC, Cer, ACar and SQDG in the GCK knockdown group, as well as remarkable downregulation in PE, TAG, DAG, GM3 and GlcADG (lipids species detected in lipidomic analysis are shown in Appendix A).The fold changes in the top 30 lipids are displayed in a matchstick plot (Appendix A).

### 2.4. GCK Knockdown Altered Hepatic Lipid Metabolism

Particularly, we analyzed the abundance of the differential metabolites involved in lipid metabolism pathways (Figure 3). Although fatty acids were not one of the top 30 lipids selected by VIP, due to their important role (the precursors of nearly all the kinds of lipids), the significantly changed fatty acids between groups were also brought into analysis. In fatty acid metabolism, palmitic acid (16:0) was drastically decreased in the GCK knockdown group; however, linoleic acid (18:2) was increased. Additionally, in ACars, the intermediates of fatty acids β-oxidation, the overall tendency was increased, despite some ACars being decreased, and correlation analysis (Figure 2D) showed that ACars were negatively correlated with TAG and DAG, suggesting active fatty acid utilization by β-oxidation in GCK-inactivating HepG2 cells. In glycerolipid metabolism, TAG and DAG were found significantly reduced, which may be due to inhibited lipogenesis. Additionally, in glycerophospholipid metabolism, most PCs showed an increasing trend, although some individual PCs were downregulated in the GCK knockdown group. The correlation analysis (Figure 2D) indicated that PCs were negatively correlated with TAG and DAG. Therefore, the biosynthesis of PC using DAG as precursors, referring to the Kennedy pathway, were promoted in HepG2 cells when GCK was inactivated. Moreover, PEs which could also be converted to PCs via *PEMT* pathways were found to be significantly reduced. In addition, sphingolipid metabolism was altered as well. GM3s were remarkably downregulated while the overall trend of Cers was upregulated in the GCK knockdown group. Taken together, these observations imply that the overall biosynthesis of PCs was preferentially enhanced, while palmitic acid, TAG and DAG were significantly reduced, probably due to inhibited synthesis or increased utilization in HepG2 cells with GCK inactivation.

### 2.5. Impact of GCK Knockdown on Lipid Metabolism-Related Enzymes and Inflammatory Genes in Human and Mouse Hepatic Cell Lines

To elucidate the mechanisms of GCK knockdown on hepatic lipid metabolism, the expression of proteins responsible for de novo lipogenesis (FASN and ACC), lipolysis (ATGL), β-oxidation of fatty acids (PPARα and CPT-1) and the Kennedy pathway for PC synthesis (CHPT-1) in the human HepG2 cells and mouse AML-12 cells was investigated using Western blot. The results showed that the levels of FASN and ACC were significantly downregulated in GCK knockdown HepG2 cells (Figure 4A). Conversely, the ratio of phosphorylated-ACC/ACC was increased (though not significant), indicating an inhibited state of ACC (Figure 4A). Additionally, the expressions of ATGL, PPARα, CPT-1 and CHPT-1 were significantly upregulated in GCK knockdown HepG2 cells (Figure 4B–D). Similar alterations in the protein levels were also found in siRNA-induced AML-12 cells, except for ATGL, which remained unaffected in both 100 nM and 200 nM dose groups (Figure 5). In the siRNA groups, the levels of GCK, FASN and ACC were reduced remarkably. Additionally, the decrease in FASN displayed a dose-dependent manner in 100 nM and 200 nM siRNA. The ratio of phosphorylated-ACC/ACC and expressions of PPARα, CPT-1 and CHPT-1 were found significantly increased. Among these, the ratio of phosphorylated-ACC/ACC was only elevated in the 200 nM siRNA-treated group. Altogether, these findings showed that GCK inactivation influenced the expression of target enzymes involved in de novo lipogenesis, lipolysis, FA oxidation and PC synthesis, resulting in reduced TAG accumulation and elevated PCs in the hepatic cells.

Additionally, we also measured the inflammatory cytokines (Figure 4E) and the expressions of NLRP3 and p-NF-kB (Appendix A) in GCK knockdown HepG2 cells under HFA challenges. The levels of IL-1β and MCP-1 were significantly decreased in GCK knockdown HepG2 cells under both normal and HFA conditions. IL-6 was significantly downregulated in GCK knockdown cells in normal conditions, but only showed a reduced trend without significance (*p* = 0.07). In the HFA group, the levels of cytokines in the control and GCK knockdown cells were both increased compared to normal groups, but the expressions of IL-1β and MCP-1 were still lower in GCK knockdown cells relative to the control. Consistently, the expressions of NLRP3 and p-NF-kB were also reduced significantly in GCK knockdown cells under normal and HFA conditions. Overall, the inflammatory markers were reduced in GCK knockdown HepG2 cells in both normal and HFA conditions, indicating the potential role of GCK knockdown in preventing inflammation induced by liptoxicity.

## 3. Discussion

Glucokinase is recognized as a glucose sensor. Recent evidence has suggested that inactivation of GCK may exert cardioprotective effects in GCK-MODY by regulating the lipid profile [7,8]. In the present study, we confirmed that GCK-MODY individuals exhibited metabolically normal and cardioprotective lipid profiles (i.e., lower TG, TC and LDLs, higher HDLs) compared to T1D and T2D. We also found that in hepatic cell models (HepG2 and AML-12), GCK inactivation improved lipid deposition and inflammation under HFA intervention and affected hepatic lipid profile by regulating key enzymes involved in lipid metabolism. These findings showed the beneficial effects of GCK inactivation in hepatic lipid metabolism and uncovered the potential mechanism of the protective lipid profile and low cardiovascular risks in GCK-MODY patients.

The liver is the major metabolic organ for lipid metabolism. We demonstrated that GCK inactivation alleviated the lipid accumulation under HFA intervention in HepG2 cells. To further investigate the molecular mechanism, lipidomic analysis was applied. A decreased level of palmitic acid as well as elevated PUFAs, including linoleic acid and docosahexaenoic acid (DHA), were detected in GCK knockdown liver cells. Evidence has shown that an excess of saturated FA palmitic acid results in lipotoxicity and inflammation in the liver, while some polyunsaturated fatty acids (PUFA), including linoleic acid, elicit opposite effects which improve insulin sensitivity and alleviate inflammation [14,15,16]. Additionally, ACars, the metric of mitochondrial β-oxidation [17], were increased in GCK knockdown HepG2 cells and were negatively correlated with DAG and TAG, suggesting an enhanced FA β-oxidation. Collectively, these results suggest that GCK inactivation may improve hepatic lipotoxicity and FA accumulation via decreasing the content of deleterious saturated FA and enhancing FA β-oxidation.

Moreover, the intracellular levels of glycerolipids TAG and DAG were both reduced in GCK knockdown HepG2 cells. Accumulating evidence has indicated that glycerolipids homeostasis is linked to glycerophospholipid [18]. When PC or PE synthesis was enhanced, the conversion of DAG to TAG would be inhibited. The main pathway for the biosynthesis of PC is the Kennedy pathway, which condenses CDP-choline with DAG to produce PC by the rate-determining enzyme cholinephosphotransferase (CHPT1) [19]. Hepatic PC synthesis has been considered metabolically beneficial in that the enhancement of PC synthesis facilitates the clearance of glycerolipids, including DAGs and TAGs, and induces the production and secretion of cardioprotective HDLs [20,21]. In this study, a correlation analysis showed that PCs were negatively correlated with TAGs and DAGs in GCK-inactivated HepG2 cells, indicating increased fluxes of lipids along the TAG-DAG-PC axis, which is in line with our previous works on serum lipidomics in GCK-MODY individuals [7]. Additionally, PEs which could be converted to PC by the PEMT pathway were found significantly reduced in the GCK knockdown group. Additionally, a reduced hepatic PC/PE ratio has been reported to be associated with hepatic steatosis, inflammation and fibrosis [22,23]. Subsequently, ELISA results showed that the levels of inflammatory cytokines (IL-1β, IL-6 and MCP-1) were significantly decreased in GCK knockdown groups under normal or HFA conditions, suggesting an anti-inflammatory state in GCK knockdown HepG2 cells. Taken together, GCK inactivation exerts favorable hepatic and serum lipid profiles, probably by promoting the biosynthesis of hepatic PC, thus inducing the clearance of TAG and DAG, increasing overall circulating HDLs and preventing liver inflammation and lipid accumulation.

Furthermore, two kinds of sphingolipids displayed significant but opposite changes in GCK knockdown HepG2 cells. The levels of several Cers were elevated, while the overall expressions of GM3 were reduced in the GCK knockdown group. It is notable that various sphingolipids were associated with lipotoxicity and inflammation, and were elevated in animal and human NAFLD and diabetes [24,25,26]. In this study, the elevation in Cers and the reduction in GM3 in hepatocytes may offset each other’s effects on lipotoxicity and inflammation, which explains the lower overall inflammation level in GCK inactivation cells.

In addition, several key enzymes involved in lipid metabolic pathways were examined in HepG2 and AML-12 cell lines. FASN and ACC are rate-limiting enzymes responsible for de novo lipogenesis in the liver [27,28]. Hepatic FASN and ACC proteins were decreased in the GCK knockdown group, which might inhibit de novo lipogenesis, thus contributing to reduced palmitic acids and TAG levels. PPARα and its downstream CPT-1 promoted FA β-oxidation in the liver [29,30]. Herein, GCK inactivation significantly upregulated the expression of PPARα and CPT-1. Additionally, ATGL and CHPT-1,which catalyzes the hydrolysis of TAG into DAG [31] and mediates PC synthesis from DAG precursors [32], were found both upregulated in GCK knockdown HepG2 cells, thus inducing lipolysis of TAG and facilitating the downstream biosynthesis of PCs. These enzymes were also examined in siRNA-induced mouse AML-12 cells, and similar alterations in the protein levels were found, except for ATGL, which remained unaffected in low- and high-dose groups, which probably suggested distinct metabolism regulations between human and mouse [33], and further investigation on animal models is needed.

This study also has several limitations. In this study, we used in vitro cell models to investigate the role of GCK inactivation in liver lipid metabolism; therefore, further in vivo experiments are needed. In addition to the liver, the effects of GCK inactivation on lipid metabolism should also be validated in other tissues, including serum and adipose tissue. Additionally, this study focuses on the partial inactivation of GCK; as there are more than 600 loss-of-function mutations of GCK [34], further studies on GCK point-mutation in cells or animal models are needed to determine whether the findings of the present study are common to different mutations of GCK.

In conclusion, partial inactivation of GCK ameliorated hepatic lipid accumulation and inflammation by altering the expressions of hepatic genes involved in lipogenesis, lipolysis and β-oxidation in HepG2 and AML-12 cell models. This finding proved that reduced GCK activity optimized hepatic lipid metabolism, providing a novel mechanism for the favorable lipid profile and low cardiovascular risks in GCK-MODY patients. Glucokinase inactivation could be a potential strategy for the prevention of diabetes-related vascular complications. Further investigations are required to explore the protective and curative effects of GCK inactivation on dyslipidemia and cardiovascular complications in diabetic and nondiabetic populations.

## 4. Materials and Methods

### 4.1. Study Population and Data Collection

This study cohort comprises GCK-MODY (n = 33), T1D (n = 34), T2D (n = 34) and healthy individuals (n = 30). All participants were recruited from the outpatient clinic and inpatient ward of the endocrinology department at the Peking Union Medical College Hospital (PUMCH), Beijing, China, between January 2017 and December 2021. Demographic information and laboratory tests were collected. The study protocol was approved by the ethical standards of the Peking Union Medical College Hospital Ethics Committee and written consent was provided from all participants.

### 4.2. Cell Culture and High Fatty Acid Treatment

Human hepatocellular carcinoma cell line (HepG2) and alpha mouse liver 12 cell line (AML-12) were obtained from the Cell Bank of the Chinese Academy of Sciences (Shanghai, China). The HepG2 cells were cultured in DMEM medium with 10% fetal bovine serum (FBS) and 1% penicillin–streptomycin (100 U/mL penicillin 10 ug/mL streptomycin) and maintained at 37 °C with 5% CO_2_. The AML-12 cells were grown in DMEM/F12 (1:1) supplemented with 0.005 mg/mL insulin, 0.005 mg/mL transferrin, 0.005 mg/mL selenium, 10% FBS and 40 ng/mL dexamethasone (Gibco, San Diego, CA, USA) at 37 °C with 5% CO_2_. For high fatty acid (HFA) treatment, cells were incubated with oleic and palmitic acid in the ratio 2:1 (500 μM/250 μM) or vehicle for 24 h or 48 h.

### 4.3. Lentivirus Transfection

Lentivirus-mediated GCK knockdown (hU6-GCK-CBh-gcGFP-IRES-puro, GV493) and control constructs were synthesized by Genechem (Shanghai, China). The GCK-KD group was transfected with GCK knockdown lentivirus, and the control group was transfected with empty lentivirus. HepG2 cells were cultured in 6-well plates (1 × 10^6^/well). When the confluency reached about 60% (24 h), HepG2 cells were transfected with the constructed human GCK knockdown lentivirus or GFP-expressing control vector (Genechem, Shanghai, China) at a multiplicity of infection (MOI) of 10, with 40 μL/mL infection enhancer HitransG P (Genechem, Shanghai, China) in the medium. The medium containing lentivirus was replaced with fresh medium after 12–16 h. Subsequently, the GCK-KD group were selected with 2 μg/mL puromycin for 72 h.

### 4.4. SiRNA Transfection

For the transient knockdown of GCK in AML-12 cells, mouse GCK-siRNA (GCTCAGAAGTTGGAGACTT) and negative control-siRNA were designed and synthesized by RIBOBIO Co., Ltd. (Guangzhou, China). The transfection of siRNA was facilitated by Lipofectamine 2000 reagent (Invitrogen, Waltham, MA, USA) with siRNA: Lipo2000 = 100 pmol:5 μL for each well of 6-well plates. Two concentration gradients of GCK-siRNA were used [100 nM (200 pmol siRNA) and 200 nM (400 pmol siRNA)] and treated for 24 h. Transfection efficiencies of GCK were confirmed with Western blot and enzyme activity examination.

### 4.5. Oil Red O Staining and Intracellular TAG Levels

The HepG2 cells were plated in 6-well plates. After the cells were fused to 40–60%, the HepG2 cells were treated with FFA as described above. After 48 h, the cells were washed three times with PBS and fixed with 4% paraformaldehyde for 30 min at room temperature. The fixed cells were washed gently with PBS and immersed in 60% isopropanol for 5 min. Then, we removed the isopropanol and stained the cell with Oil Red O solution for 20 min and Mayer’s Hematoxylin Stain solution for 1 min. The excess dye was removed, and the cells were washed four times with distilled water before the microscopic observation under the bright field.

For intracellular TAG levels, after treatment with FFA for 48 h, the HepG2 cells were harvested and lysed to prepare cell lysates. Intracellular TG levels were measured using a triglyceride quantification kit (MICHY Biology, Suzhou, China) following the manufacturer’s instructions. The protein contents in the lysate were determined using the bicinchoninic acid kit (Invitrogen, Waltham, MA, USA). The output optical density was read immediately using a microplate reader at the wavelength of 505 nm. The TG content was measured as mg/mg protein.

### 4.6. Lipidomic Analysis

The lipid profiling in GCK knockdown HepG2 cells was further investigated using lipidomic analysis. As previously reported [35,36], for each sample, 480 μL of extracting solution (MTBE: MEOH = 5:1) was added for metabolites extraction. The samples were centrifuged, and the supernatants were analyzed by LC/MS. LC-MS/MS analyses were performed using an UHPLC system (Vanquish, Thermo Fisher Scientific) with a UPLC HSS T3 column (2.1 mm × 100 mm, 1.8 μm) coupled to Q Exactive HFX mass spectrometer (Orbitrap MS, Thermo) in both positive and negative electrospray ionization models. The QE mass spectrometer was used for its ability to acquire MS/MS spectra on data-dependent acquisition (DDA) mode in the control of the acquisition software (Xcalibur 4.0.27, Thermo).

### 4.7. Data Processing

The raw data files were converted to files in mzXML format using the ‘msconvert’ program from ProteoWizard. The CentWave algorithm in XCMS [37] was used for peak detection, extraction, alignment and integration, the minfrac for annotation was set at 0.5 and the cutoff for annotation was set at 0.3. Lipid identification was achieved through a spectral match using LipidBlast library, which was developed using R and based on XCMS. Orthogonal projections to latent structures discriminant analysis (OPLS-DA) were performed to identify the source of variation between groups. A permutation test repeated 200 times was conducted to ensure the model without overfitting. Variable importance for the projection (VIP) values exceeding 1.0 and *p* values of Kruskal–Wallis tests or Student’s t test (*p* < 0.05) were selected as discriminated metabolites. Correlations between lipids were analyzed by Pearson’s correlation. For the meta-analysis, the peak intensity data were converted using the Z-score transformation to represent metabolite abundance.

### 4.8. GCK Enzyme Activity Determination

The enzyme activities of GCK in cells were assessed by using commercial assay kits (Abcam, Cambridge, UK) according to the manufacturers’ instructions. GCK converts glucose into glucose-6-phosphate and produces a series of intermediates (NADPH) which could be detected by the probe, generating an intense fluorescence product (Ex/Em = 535/587 nm). Briefly, the cell lysate was diluted by GCK assay buffer (Tris-HCl buffer, pH 8.0). The reaction medium included Tris-HCl, pH 7.4, MgCl_2_, dithiothreitol, 0.1% bovine serum albumin, KCl, glucose, nicotinamide adenine dinucleotide phosphate, glucose-6-phosphate dehydrogenase and probe for NADPH. The definition of one unit of glucokinase activity is the amount of enzyme that catalyzes the release of 1.0 µmol of NADPH per minute at pH 8.0 and room temperature. The fluorescence was measured by a microplate reader (Biotek SynergyNeo2, BioTek, Vermont, VT, USA).

### 4.9. Enzyme-Linked Immunosorbent Assay (ELISA)

The supernatant of control and GCK knockdown cells was harvested after 48 h of HFA treatment and stored at −80 °C after centrifugation. The levels of Interleukin 1β (IL-1β), Interleukin 6 (IL-6) and monocyte chemotactic protein 1 (MCP-1) in supernatant were detected by ELISA kits (MULTI Science, Hangzhou, China) according to the manufacturers’ protocols.

### 4.10. Western Blotting

Cultured cells were harvested and lysed with RIPA containing 1% PMSF and phosphatase inhibitors for 30 min. Total protein concentrations were determined by the bicinchoninic acid kit (Invitrogen, USA) according to the manufacturer’s instructions. Equal amounts of total protein lysates were separated by SDS-PAGE and transferred to a PVDF membrane and blocked with 5% nonfat dry milk. The membranes were incubated overnight at 4 °C with the primary antibodies (Abcam, Cambridge, UK). The membranes were washed with TBST and incubated with an HRP-conjugated secondary antibody for 90 min. The blots were visualized using an enhanced chemiluminescence detection system.

### 4.11. Statistical Analysis

Statistical analysis was performed using the GraphPad Prism software (version 8.0.2, San Diego, CA, USA). Continuous variables were presented as mean ± SD or median (interquartile range), as appropriate. For two groups, an unpaired two-tailed t-test was performed for intergroup comparisons. For more than two groups, one-way analysis of variance (ANOVA) was used to assess statistically significant differences. Differences between groups were considered statistically significant when *p* values ≤ 0.05.

## Figures and Tables

**Figure 1 ijms-24-04315-f001:**
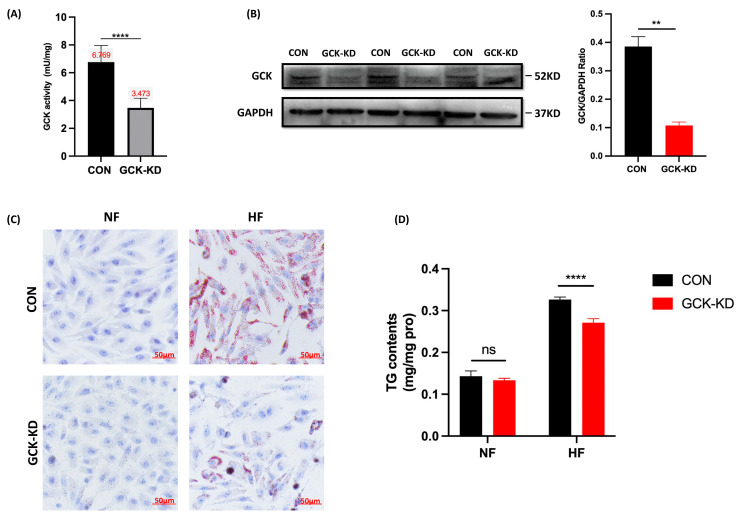
The effect of GCK knockdown on lipid accumulation and triacylglycerol content under high-fat intervention in liver cells. (**A**,**B**) Establishment of GCK knockdown human liver cell line. (**A**) Level of glucokinase enzyme activity in HepG2 cell after GCK knockdown (n = 5); (**B**) GCK protein expression level was analyzed by Western Blot in GCK knockdown liver cell (n = 4); (**C**) Intracellular lipid droplets stained by Oil Red O and triacylglycerol content (n = 3); (**D**) GCK knockdown liver cells after high-fat treatment (oleic and palmitic acid in the ratio 2:1) (n = 3). Data are represented as mean ± SD. **** *p* ≤ 0.0001, ** *p* ≤ 0.01, ns non-significant versus control group via Student’s *t*-test. CON, control group; GCK-KD, GCK knockdown group; NF, vehicle; HF, high-FFA treatment.

**Figure 2 ijms-24-04315-f002:**
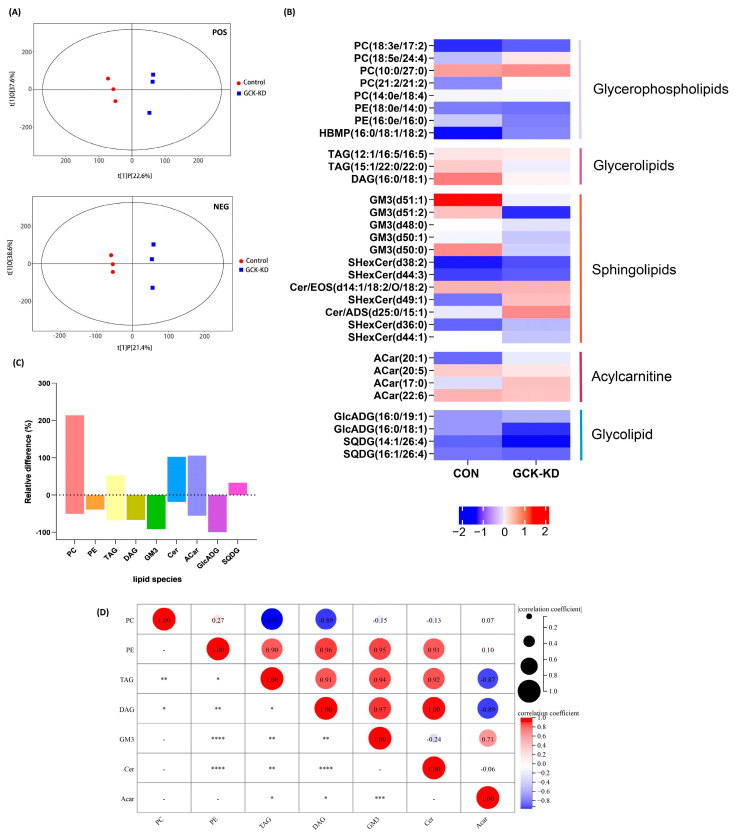
Lipidome profile of HepG2 hepatocytes after GCK knockdown. (**A**) Score scatter plot of OPLS-DA model for groups in negative and positive ionization modes; the R2Y and Q2 of the positive mode were 0.984 and 0.703, and 0.999 and 0.71 in the negative mode, respectively, indicating a high validity and good predictive ability of the model. (**B**) Heatmap analysis of the significantly altered lipid metabolites in GCK knockdown liver cells. (**C**) Bar plot for the relative change (percentage) of lipid species between groups. (**D**) Correlation matrices of lipids in GCK knockdown and control group. Sizes and colors of circle indicate correlation coefficients (positive correlation: red; negative correlation: blue). (n = 3) **** *p* ≤ 0.0001, *** *p* ≤ 0.001, ** *p* ≤ 0.01, * *p* ≤ 0.05 versus control group. Acar, Acylcarnitine; Cer, ceramides; DAG, Diacylglycerol; FA, Free fatty acid; GlcADG, glucuronosyldiacylglycerol; GM3, Ganglioside; PC, Phosphatidylcholine; PE, Phosphatidylethanolamine; SQDG, Sulfoquinovosyl diacylglycerol; TAG, triacylglycerol.

**Figure 3 ijms-24-04315-f003:**
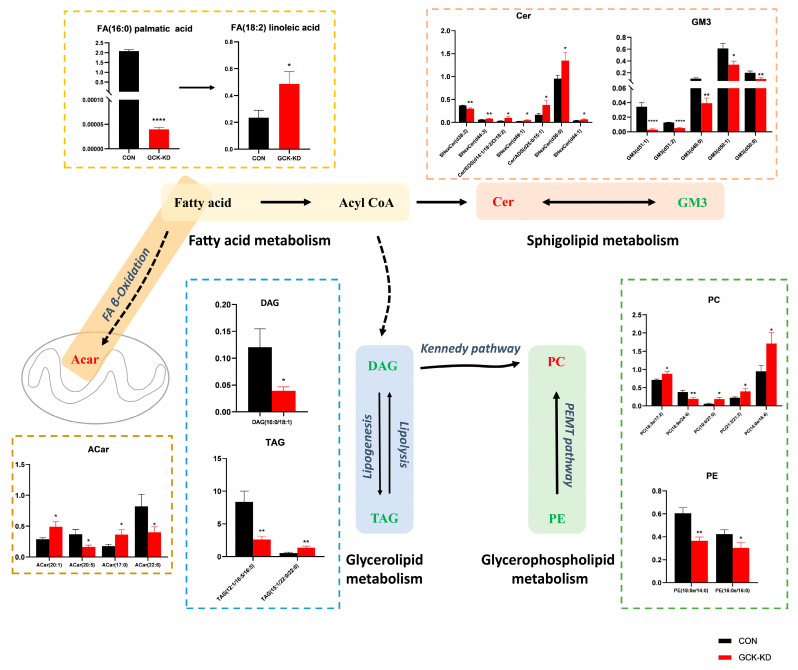
Alterations of lipid metabolism in GCK knockdown HepG2 cells. Lipid metabolites involved in fatty acid synthesis, glycerolipid metabolism, acylcarnitine metabolism, glycerophospholipid metabolism and sphingolipid metabolism were changed in hepatocytes after GCK knockdown. The *Y*-axis displays relative quantitative value of each lipid. Data are represented as mean ± SD. (n = 3) **** *p* ≤ 0.0001, ** *p* ≤ 0.01, * *p* ≤ 0.05 versus control group via Student’s *t*-test. The lipid metabolites in red represent upregulation, and those in green indicate downregulation. Acar, Acylcarnitine; Cer, Ceramides; DAG, Diacylglycerol; FA, Free fatty acid; GM3, Ganglioside; PC, Phosphatidylcholine; PE, Phosphatidylethanolamine; TAG, triacylglycerol.

**Figure 4 ijms-24-04315-f004:**
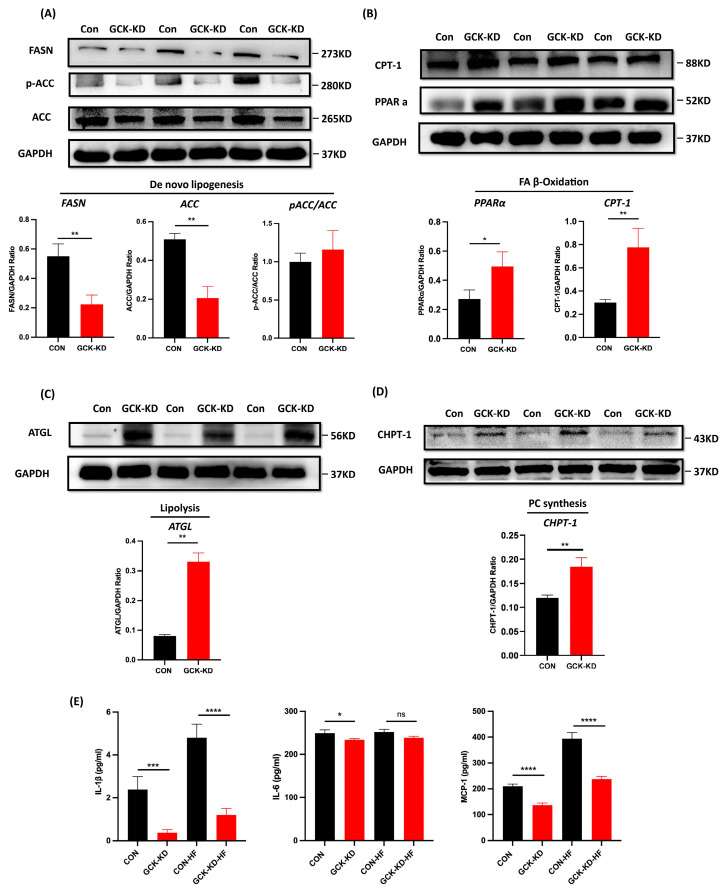
GCK knockdown impacts the expression of proteins involved in de novo lipogenesis, lipolysis, fatty acid β-oxidation, phosphatidylcholine synthesis and inflammation in HepG2 cells. (**A**) Protein expression of FASN, ACC and phosphorylated-ACC in GCK knockdown HepG2 cells. Protein levels normalized to GAPDH protein content are shown as ratio. Phosphorylated-ACC protein levels are normalized to total ACC. Data represent the mean ± SD. A representative blot is shown on the top. (**B**) Relative expression of CPT-1 and PPARα protein in GCK knockdown HepG2 cells. Normalized ATGL (**C**) and GHPT-1 (**D**) protein expression are analyzed in in GCK knockdown HepG2 cells. Representative blots are shown on the top. Protein levels normalized to GAPDH protein content are shown as ratio. (**E**) The levels of cytokines IL-1β, IL-6 and MCP-1 in supernatant of GCK knockdown HepG2 cells under HFA intervention were analyzed by ELISA. Data are represented as mean ± SD. (n = 3) **** *p* ≤ 0.0001, *** *p* ≤ 0.001, ** *p* ≤ 0.01, * *p* ≤ 0.05, ns non-significant versus control group via Student’s *t*-test. FASN, fatty acid synthase; ACC, acetyl-CoA carboxylase; CPT-1, carnitine palmitoyltransferase-1; PPARα, peroxisome proliferator-activated receptor α; ATGL, adipose triglyceride lipase; CHPT-1, Cholinephosphotransferase-1; IL-1β, Interleukin 1β; IL-6, Interleukin 6; MCP-1, monocyte chemotactic protein 1; NF, vehicle; HFA, high fatty acid treatment.

**Figure 5 ijms-24-04315-f005:**
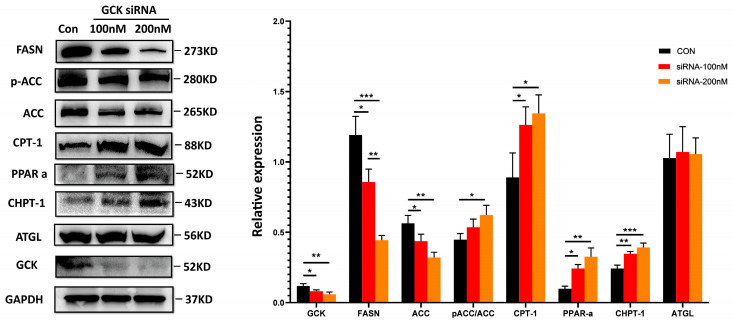
siRNA-induced GCK suppression in mouse AML-12 cell influences the expression of proteins involved in de novo lipogenesis, fatty acid β-oxidation and phosphatidylcholine synthesis. Western blots of proteins including FASN, ACC, p-ACC, CPT-1, PPARα, ATGL and GCK in AML-12 cell treated with vehicle, 100 nM and 200 nM GCK-siRNA are shown on the left. Protein levels normalized to GAPDH protein content are shown as ratio on the right. Data are represented as mean ± SD. (n = 3) *** *p* ≤ 0.001, ** *p* ≤ 0.01, * *p* ≤ 0.05 via one-way ANOVA test.

**Table 1 ijms-24-04315-t001:** Demographic and lipid profile of the diabetic subjects.

Parameters	MODY2 (n = 33)	T1DM (n = 34)	T2DM (n = 34)	Control (n = 30)	*p*
Gender (M/F)	12/21	14/20	13/21	14/16	0.853
Age (Year)	25.4 ± 16.42	18.6 ± 9.54	24.7 ± 7.91	16.8 ± 10.14 &	**<0.01**
BMI (kg/m^2^)	19.1 ± 4.44	19.0 ± 4.97	25.7 ± 4.53 *	19.6 ± 3.11	**<0.0001**
FBG (mmol/L)	6.7 (6.4, 7.1)	7.8 (6.4, 10.5) #	7.6 (6.4, 9.9) *	4.7 (4.5, 5.1) &	**<0.0001**
Fasting insulin (mIU/L)	5.4 (3.9, 8.9)	16.4 (10.6, 24.2) #	13.7 (8.1, 21.8)	7.9 (4.9, 10.2)	**0.0019**
Fasting C-peptide (ng/mL)	0.89 (0.72, 1.36)	0.37 (0.18, 0.52) #	1.36 (0.85, 2.02) *	1.13 (0.71, 1.25)	**<0.0001**
HbA1c (%)	6.3 ± 0.37	8.7 ± 2.29 #	8.4 ± 2.3 *	4.7 ± 0.4 &	**<0.001**
GA (%)	18.0 ± 1.46	23.5 ± 6.39 #	22.3 ± 6.37 *	13.6 ± 1.19 &	**<0.0001**
TAG (mmol/L)	0.47 (0.39, 0.86)	0.64 (0.46, 0.81)	1.59 (1.36, 1.8) *	0.73 (0.48, 1.02)	**<0.0001**
TC (mmol/L)	4.4 ± 0.77	4.5 ± 0.86	5.3 ± 1.05 *	4.0 ± 0.67	**<0.0001**
LDL-C (mmol/L)	2.1 ± 0.57	2.6 ± 0.76	3.3 ± 1.01 *	2.2 ± 0.68	**<0.0001**
HDL-C (mmol/L)	1.6 ± 0.25	1.3 ± 0.23 #	1.2 ± 0.42 *	1.5 ± 0.21	**<0.0001**
ALT (mmol/L)	14 (11, 18)	14 (10, 18.5)	21 (16, 33) *	11 (9, 15.5)	**<0.0001**
Cr (umol/L)	55.9 ± 18.10	52.6 ± 16.64	60.8 ± 26.3	53.6 ± 11.32	0.3371
UA (umol/L)	258.7 ± 73.32	255.6 ± 64.20	362.3 ± 109. 9 *	307.5 ± 93.07	**<0.0001**
CRP (mg/L)	0.19 (0.16, 0.36)	0.66 (0.28, 1.58) #	1.3 (0.49, 1.95) *	0.26 (0.17, 0.44)	**0.0065**
Hcy (umol/L)	9.3 (8.5, 11.1)	9.7 (8, 10.3)	16.1 (10.1, 17.1)	11 (9.5, 13.8)	**0.0087**

The data were expressed as mean ± standard deviation after one-way ANOVA; Non-normally distributed data were expressed as the median (25th–75th) after independent sample nonparametric test. *p* < 0.05 indicates a significant difference, which is indicated in bold; Differences between MODY2 and T2DM are indicated by *, differences from T1DM are indicated by # and differences from control are indicated by &. BMI: body mass index; FBG: fasting blood glucose; HbA1c: glycated hemoglobin; GA: glycated albumin; TAG: triacylglycerol; TC: total cholesterol; LDL: low-density lipoproteins; HDL: high-density lipoproteins; Cr: creatinine; UA: uric acid; CRP: C reactive protein; Hcy: Homocysteine.

## Data Availability

The data that support the findings of this study are available from the corresponding author upon reasonable request. Individual-level data cannot be shared for reasons of patient privacy.

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
