# Peer review of "Glucokinase Inactivation Ameliorates Lipid Accumulation and Exerts Favorable Effects on Lipid Metabolism in Hepatocytes"

_ijms, 2023, doi:10.3390/ijms24054315_

Round 1

Reviewer 1 Report

In the present study, the authors evaluate the impact of GCK inactivation on hepatic lipid metabolism and inflammation in order to better understand the low incidence of vascular complications observed in GCK-MODY patients, when compared to patients with other forms of diabetes. The manuscript is well structured and presented, but it has some weakness which are acknowledged by the authors in the discussion section.

I have a minor suggestion concerning the validation of the functional impact of GCK downregulation: the authors address the limitation of using in vitro cell models to investigate the role of GCK-inactivation. It would be interesting if the downregulation of FASN and ACC enzymes could be validated in peripheral blood samples from the studied MODY2 patients. Or is this a tissue specific effect?

Another suggestion that could reinforce the impact of GCK on the enzymes related to lipid metabolic pathways, would be to test if there is any correlation between GCK gene expression and the different enzymes, on liver, and pancreas, using data from publicly available datasets such as Genotype-Tissue Expression (GTEx) database.

Reviewer 2 Report

The manuscript describes changes in the lipid metabolism by glucokinase inactivation by a MODY mutation either under in vitro conditions in the HepG2  and AML12 cell lines or in vivo using cohort data from GCK-MODY, T1D and T2D individuals compared to non-diabetic subjects.

In the materials and methods section are the non-diabetic individuals healthy or in respect to the lipid profile?

For the HFA treatment, the authors used a specific ratio of PA and OA. Is this used ratio their own experience or from the literature?

In the result section, did the author mean lipid or lipid droplets accumulation in the HepG2 cell line?

The used abbreviations cannot be explained in the Suppl. Table.  This is a disadvantage for the reader.

The discussion section is too long.

The limitations of the present study are good, but not at the end of the MS.

At this place, please add a take home message.

Author Response

We thank you for the time and effort that you have put into reviewing the previous version of the manuscript. Your suggestions have enabled us to improve our work. We agree with these comments, and we have addressed these specific concerns point by point. The ‘Track changes’ feature in Microsoft Word and all changes made are easily identifiable in our paper.

The manuscript describes changes in the lipid metabolism by glucokinase inactivation by a MODY mutation either under in vitro conditions in the HepG2 and AML12 cell lines or in vivo using cohort data from GCK-MODY, T1D and T2D individuals compared to non-diabetic subjects.

-In the materials and methods section are the non-diabetic individuals healthy or in respect to the lipid profile?

Answer: The healthy controls enrolled in our study are healthy and without any disorders including diabetes, cardiovascular diseases, NAFLD, cancer, etc. Besides, who drink alcohol was also excluded in order to eliminate the effects of alcohol on the liver.

-For the HFA treatment, the authors used a specific ratio of PA and OA. Is this used ratio their own experience or from the literature?

Answer: Yes, the ratio of PA/OA (1:2) used in this study for inducing fatty liver model was based on the literatures [1, 2]. In the previous studies, HepG2 cells and primary hepatocytes were treated with oleic acid and palmitic acid (O/P) (2:1, M/M) to induce hepatic steatosis and inflammation.

-In the result section, did the author mean lipid or lipid droplets accumulation in the HepG2 cell line?

Answer: We indicated lipid accumulation. Oil red O solution has been used to stain neutral lipids (triglycerides) and some lipoproteins in cells or tissues in multiple studies [3-5]. In this study, red oil O, intracellular TG content and lipidomics results showed that the main type of natural fat (triacylglycerides) was significantly reduced, suggesting improved lipid accumulation in GCK-inactivation hepatocytes. And the levels of enzymes involved in de novo lipogenesis and lipolysis also confirmed this.

-The used abbreviations cannot be explained in the Suppl. Table.  This is a disadvantage for the reader.

Answer: Thank you for your advice. The used abbreviations have been added and highlighted in the manuscript (behind the Abstract). The Suppl. Table 1 was only for the reference of lipid species from lipidomic analysis.

-The discussion section is too long.

Answer: Thank you for your advice. The discussion has been simplified, and the overlapped contents were condensed. Please see the “manuscript-Discussion” for details.

-The limitations of the present study are good, but not at the end of the MS.

At this place, please add a take home message.

Answer: Thank you for your suggestions. The location of “Limitation” has been altered and a summary of our work was added at the end of this manuscript. Please see the “manuscript-Discussion” for details.

References:

  1. Liu, H., et al., Enhanced alleviation of insulin resistance via the IRS-1/Akt/FOXO1 pathway by combining quercetin and EGCG and involving miR-27a-3p and miR-96-5p.Free Radical Biology & Medicine, 2022. 181: p. 105-117.
  2. Choi, S.-E., et al., Mitochondrial protease ClpP supplementation ameliorates diet-induced NASH in mice.Journal of Hepatology, 2022. 77(3): p. 735-747.
  3. Zhang, W., et al., Hepatic Peroxisome Proliferator-Activated Receptor Gamma Signaling Contributes to Alcohol-Induced Hepatic Steatosis and Inflammation in Mice.Alcoholism, Clinical and Experimental Research, 2016. 40(5): p. 988-999.
  4. Zhu, X., et al., Berberine attenuates nonalcoholic hepatic steatosis through the AMPK-SREBP-1c-SCD1 pathway.Free Radical Biology & Medicine, 2019. 141: p. 192-204.
  5. Jung, T.W., et al., β-aminoisobutyric acid attenuates LPS-induced inflammation and insulin resistance in adipocytes through AMPK-mediated pathway.Journal of Biomedical Science, 2018. 25(1): p. 27.
